# Analysis of the Impact of Water Flow Rate on the Temperature Variability in a Closed Room during the Extinguishing of A-Group Fire Using a Hybrid Water Mist Suppression System

**Jerzy Gałaj** [1],[*] , **Norbert Tuśnio** [2] , **Paweł Wolny** [3] and **Tomasz Drzymała** [2]

[1]  Institute of Safety Engineering, The Main School of Fire Service, Słowackiego Str. 52/54,
     01-629 Warsaw, Poland

[2]  Faculty of Safety Engineering and Civil Protection, The Main School of Fire Service, Słowackiego Str. 52/54,
     01-629 Warsaw, Poland; ntusnio@sgsp.edu.pl (N.T.); tdrzymala@sgsp.edu.pl (T.D.)

[3]  Faculty of Process Engineering and Environmental Protection, Lodz University of Technology,
     Wólczańska 213, 90-001 Łódź, Poland; pawel.wolny@p.lodz.pl

[*]  Correspondence: jgalaj@sgsp.edu.pl; Tel.: +48-693-175-252

**Abstract:** The advantage of hybrid fire suppression systems is that they combine the advantages of both water mist and clean agent systems. Currently, this innovative technology is increasingly used in fixed firefighting systems. Available literature both in Polish and around the world describes this issue in a fragmentary way. Extinguishing system tests were carried out at the Main School of Fire Service in Warsaw. The impact of water mist flow rate on the temperature variability in a closed room during the extinguishing of group A fires using a hybrid water mist system was analyzed. Four different flow rates 0.5, 1.0, 1.5 and 3.0 dm$^3$/min were applied. The temperature variability of selected points of a closed test chamber during extinguishing process are presented. The extinguishing efficiency, taking into account extinguishing time, average speed of temperature drop and other parameters, was estimated on the basis of proposed original criterion. The results obtained showed that water flow rate has a significant impact on temperature variability and included the determination of optimal water flow rate at which the extinguishing efficiency is the highest. The conducted research proved that the hybrid system is more effective than classic mist and gas systems when extinguishing group A fires.

**Keywords:** hybrid water mist system; internal fire; fire extinguishing; water mist fire suppression system; fire environment; fixed extinguishing device; group A fire

## 1. Introduction

The first mention of hybrid systems can be found in papers [1,2] in which, in addition to discussing the typical mist systems, studies of those systems in which an additional agent in the form of inert gas was used. The results of the tests conducted by Liu and Kim in 1996 showed that the combination of water mist and inert gas improves the extinguishing efficiency compared to conventional water mist systems. Although the beginnings of research related to hybrid systems took place over 20 years ago, more interest in hybrid systems has only occurred in recent years. An overview of these types of extinguishing systems developed by Raia and Gollner can be found in [3]. The tests carried out by American insurance company FM Global, which, due to the lack of appropriate standards, determine the current range of applications of hybrid systems, were also described in the work. The fire tests for hybrid extinguishing systems, in which heptane pool fires (B group fire) were applied, covered the following cases:

1.  Protection of combustion turbine in enclosures with volumes not exceeding 80 m$^3$ and 260 m$^3$;
2.  Protection of machinery in enclosures with volumes not exceeding 80 m$^3$ and 260 m$^3$;
3.  Protection of combustion turbine and machinery enclosures with volumes exceeding 260 m$^3$;
4.  Protection of computer room raised floors.

The guidelines for project implementation, as well as requirements, for all fire scenarios listed above were included in [4]. The FM 5580 approval also specifies the requirements for special fire extinguishing systems that list three different fire scenarios. The adopted assumptions set the time frame for system operation and effective firefighting. These times are: 0–5 min, 5–8 min and longer than 8 min. The minimum response time is specified for all time intervals. In the first two cases (0–5 min and 5–8 min), a minimum discharge time of 10 min is required. If the threshold of 8 min is exceeded, the device does not meet the approval requirements and as such cannot be used. The approval is issued based on the assessment of the system. The certification covers the system's effectiveness and compliance with the approval requirements and the manufacturer's declaration, as well as its durability and reliability. Maintenance of certification is only possible if tests and inspections are carried out in accordance with the manufacturer's requirements. Tests and inspections must be carried out by an authorized service center. Test specification and system evaluation criteria for each case (type of combustible material, fuel delivery method, nozzle arrangement, fire location, incubation time) are strictly specified in FM Approval 5580.

Hybrid systems have a unique scalability feature. Depending on the application, the hybrid system can be used for both local security and complete protection of the facility. The ability to adapt hybrid systems to requirements and expectations in the field of fire protection is an advantage. Currently available data suggest the feasibility of using a hybrid system for comprehensive protection of facilities as an alternative solution to conventional sprinkler systems. The hybrid system used for comprehensive building protection is to operate as defined in [4] for indoor fire tests. Inert gas and water mist are to extinguish fires synergistically. In this solution, none of the extinguishing agents used is able to extinguish the fire on its own within a given time. The application is more often used where the hybrid system works more as a water mist system. Depending on the solution adopted, the proportion of water mist and gas changes, which means that the components of the hybrid system no longer need to affect the fire environment evenly to put out the fire. If such a system is more dependent on water mist or inert gas as a leading extinguishing agent for firefighting, it may exhibit mist or inert gas characteristics that deviate from the basic properties of the hybrid system. It is possible to configure a hybrid system so that the proportion of extinguishing media is changed automatically by measuring the oxygen concentration in the fire environment.

Based on the research carried out by FM Global described in [4], hybrid systems have been included in a separate category. Despite the similarities to traditional inert gas systems and water mist, there are clearly differences that must be taken into account in the design process of installation, servicing and inspection of hybrid fire extinguishing systems. Two options for classifying hybrid systems were considered. The first scenario was to incorporate hybrid technology into existing NFPA 2001 or NFPA 750 standards [5,6]. The second scenario was to create a new separate standard for hybrid systems. Despite the similarity, adapting hybrid fire extinguishing systems to both NFPA 2001 and 750 requires a different specific certification method. This is due to the fact that the NFPA 2001 standard explicitly excludes systems that use water as the primary extinguishing agent, which automatically excludes from this standard certification of certain hybrid systems configurations where water fog is the dominant factor. The NFPA Technical Committee began work on a document dedicated to hybrid systems, appearing in the NFPA classification under number 770, which was to be completed in April 2020 [7].

From the hybrid systems currently on the market, more information is available only for two solutions [8]. One of them is the Aquasonic® Water-Atomizing Fire Suppression system designed and produced by ANSUL, currently part of the Tyco group. In Europe, this system is manufactured by Siemens (Sinorix H$_2$O Gas system). The Aquasonic® system is used for fire protection intended

for group B fires. It is a specially designed, effective fire protection system used to protect technical rooms and spaces inside machine housings such as combustion turbines, machine compartments, generator housings and flammable liquid storage. The system uses non-toxic and readily available extinguishing agents, which include water and nitrogen. The kit includes two supersonic nozzles that produce about 1.5 trillion drops of water per second with a total area of 121 $m^2$ per second. This surface is closely related to heat dissipation from the fire environment by changing the liquid water phase into water vapor. Thanks to the high speed given to the droplets by the nozzles, the water mist is able to reach the entire combustion zone. According to the manufacturer's assurance, the Aquasonic® hybrid nozzles are made of stainless steel and do not contain internal moving parts, which guarantees their high strength and reliability. The Aquasonic® system uses water drops as the extinguishing agent. This is due to the need to comply with the requirements of the NFPA 750 standard, which does not take into account extinguishing agents other than water mist. When the NFPA 770 standard is developed, system re-certification is likely to occur. The fire extinguishing system is activated after a fire is detected in the protected area by the detection and control system of the Aquasonic® system. The fire set consists of a nitrogen cylinder and a water tank made of stainless steel. The pressure from the discharge of nitrogen drives the water stream to the mist nozzles. The activation of the system opens the valves on the nitrogen cylinders and the gas under full pressure flows through the regulators, maintaining a constant pressure of 8.6 bar. This pressure forces water to flow through the open valve and nozzles. Discharge of nitrogen also provides the necessary energy to create droplets of water mist. Manufacturers of water systems indicate that they are not used to directly extinguish materials that react in contact with water because of the possibility of violent reactions or significant amounts of hazardous gases.

A competitive product classified as a hybrid system is Vortex®, designed and produced by Victaulic. In Europe, this system is produced by the Italian company Tema Sistemi (Aquatech® Water Mist System). According to the manufacturer's declaration, it uses only about 1.5 $dm^3$ of water for effective extinguishing, due to the proportions of both extinguishing agents used. There is the possibility of any adjustment of the percentage of content in the water–gas system. For installations with a small range of impacts, nitrogen is used as a lead. It displaces oxygen from the fire environment, while water fog has a cooling effect. For larger protected areas, water mist acts as the primary extinguishing agent by absorbing heat and displacing oxygen in the process of rapid conversion into water vapor. By using inert gases, some of the hybrid systems produce droplets of the order of 10 μm. The diameter of classic water mist droplets is on average 10 times larger. This results in better heat absorption, resulting from a much larger evaporation surface, which increases the extinguishing efficiency. There are a few differences between Aquasonic® and Vortex® systems. The first difference in technical parameters is different working pressures on the nozzles. In the Vortex® system, nitrogen is fed under a pressure of 1.7 bar and water, 2 bar. In turn, the Aquasonic® system uses a higher nitrogen pressure of 4–7 bar, and water in the range of 2–4 bar. The Vortex® system has passed the tests resulting from the requirements of FM Global 5560 approval for closed rooms with a volume of 130 and 260 $m^3$ (2004) and a volume of 550 $m^3$ (2005). The certification also covered the protection of the interior of machinery and equipment as well as for flammable and extremely flammable substances using two and four heads depending on the protected surface. In 2006 and 2007, FM Global 5560 approval was obtained for systems intended for the protection of rooms with a volume of 880 to 3600 $m^3$ [9]. In 2009, two types of fire extinguishing systems were launched—Victaulic 1500 and Victaulic 1000. The first requires an individual design adapted to the specificity of the protected area. The method of operation in a fire environment is to reduce the oxygen concentration to 14–12.5% $O_2$. The design discharge time is 3–5 min. The second set is designed to protect the interior of steam or combustion turbines as well as other machinery and equipment. The design discharge time is 10 min.

The combination of water mist and inert gases eliminates some of their imperfections. In the case of low-pressure mist, it is air powered, which oxygenates the fire (this only affects permanent fire extinguishing devices located in rooms), while the disadvantages of gas systems are the high cost of

installation and the need to ensure adequate tightness of the extinguished room, the use of technical solutions, such as relief flaps, and the risk of damage to protected property due to the rapid flow of extinguishing gas.

Many fire tests connected with the extinguishing efficiency of streams produced by selected mist nozzles when firefighting type A fires in a closed room were conducted in the Main School of Fire Service. The research consisted in recording the temperature and toxic gas concentrations at selected points of the room when extinguishing burning wood. The measurement of parameters lasted from the moment of fire initiation to its complete extinguishing. Obtained temperature during extinguishing will allow for the validation of extinguishing models developed at the Main School of Fire Service in which the mist nozzle was used as extinguishing equipment [10–16]. Experiments related to fire suppression with the help of hybrid systems were also performed. Their main purpose was to analyze the impact of the water mist flow rate and the type of gas used in the hybrid fire extinguishing system on the concentration of toxic gases and oxygen in a closed room during the extinguishing of group A fires. During the tests, it was clearly demonstrated that the gas used in the water mist fire suppression system it is not only a driving factor, but also participates in the extinguishing process. The experiment showed that compared to classic mist and gas systems, the hybrid system is more effective because it combines the advantages of fire suppression with water mist, which causes a rapid reduction of temperature in the combustion zone by receiving a large amount of heat and fire suppression with inert gas, which reduces oxygen concentration and eliminates free radicals [17–19].

The objective of this work was the analysis of the impact of flow rate of nitrogen-powered water mist in a specially designed and constructed hybrid system on the variability of temperature in a closed room during a wood fire. The amount of water supplied has an effect on the extinguishing efficiency, since a larger number of drops ensures more intense heat removal. However, too much of them may no longer significantly improve the extinguishing efficiency, increasing its consumption and fire losses. Accordingly, this study has, among other things, specified what should be the amount of water supplied in relation to the amount of gas supplied for the assumed fire scenario. These tests were particularly important due to the fact that, to date, this type of fire test has not been carried out using hybrid systems. In addition, a new intuitive mathematical criterion for extinguishing effectiveness has been proposed for the first time. On the basis of the value of proposed criterion, it will be possible to assess the extinguishing effectiveness of any fire extinguishing system, including the hybrid system. The results obtained can be useful in developing a new standard that would include hybrid fire extinguishing systems.

## 2. Materials and Methods

*Experimental Setup*

A hybrid fire extinguishing system including four two-phase atomizing nozzles FEN T was the subject of the study. Technical parameters of the nozzles are given in Table 1. The view of one of the nozzles is shown in Figure 1. Inside the nozzle, a pin is mounted axially, through which liquid is fed into the gas stream, behind the nozzle's throat. The ratio of volumetric flow rate of both media (for the volume of expanded gas to atmospheric pressure) for the FEN T series is 500:1. The arrangement of two 110.5 FEN T nozzles and two 110.1 FEN T nozzles was assembled on a two-part supply manifold, allowing a significant change in the orientation of each mist stream. The effectiveness of water extinguishing fog is considered according to the scheme:

- absorption of thermal energy (cooling);
- displacing oxygen from the atmosphere;
- creating a screen protecting against heat energy radiation;
- isolation from a fire source by propagating a mixture of steam and air;
- kinetic effects.



**Table 1.** Technical data of the nozzle FEN T.

| Technical Parameter | Gas | Water |
|---|---|---|
| Input pressure | 4 ± 0.5 bar | 4 ± 0.5 bar |
| Nominal flow rate | 1 ± 0.2/0.5 ± 0.1 m³/min * | 2 ± 0.2/1 ± 0.1 dm³/min * |
| Media cleanliness | solid particles d < 40 μm particle density < 10 mg/m³ | filter 300 μm |
| Power connections | external thread 1/4" | external thread 1/8" |
| Head mass | 0.4 kg | |
| Effective range of the stream | 2.5 m | |
| Maximum range of the stream | 3.5 m | |
| Operating temperature | from 10 to 700 °C | |
| Droplet size | from 4 to 200 μm | |
| Mean volumetric droplet size | from 20 to 30 μm | |
| Mounting | threaded hole M6 | |

* The first value is for the nozzle 110.1 and the second for the nozzle 110.5. The other parameters are the same for both nozzles.

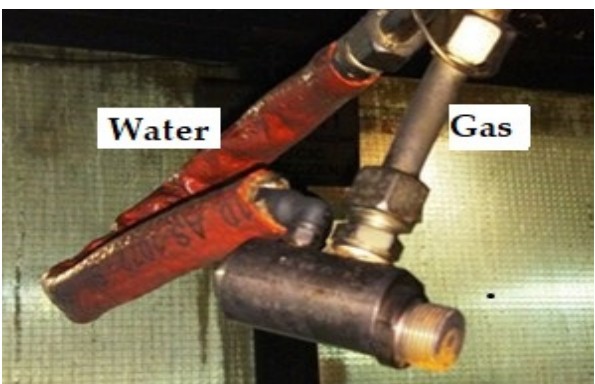

**Figure 1.** A view of the two-phase nozzle.

The tests were carried out in a closed room measuring 5 × 5 × 2.8 m, in which two walls were lined with ceramic tiles and two were made of tempered glass resistant to high temperature. Water outflow was provided by a drainage channel located in the center of the room. The installation was attached to the frame at a distance of 40 cm from the ceiling. The water line was fed using a pump set located in an adjacent room. The gas installation was supplied from a bundle of cylinders connected in parallel, in which there was nitrogen at a pressure of 200 bar. Then gas was supplied to the reducer via a high-pressure line. The output pressure of the reducer was 4 bar. Water and gas installations were made of steel pipes connected by a Parker system. It involved the use of pipe fittings for connections or connecting rigid and flexible pipes. The extinguishing system diagram is shown in Figure 2. The blue line indicates the lines through which the water is supplied and the green line shows the lines through which the gas is supplied.

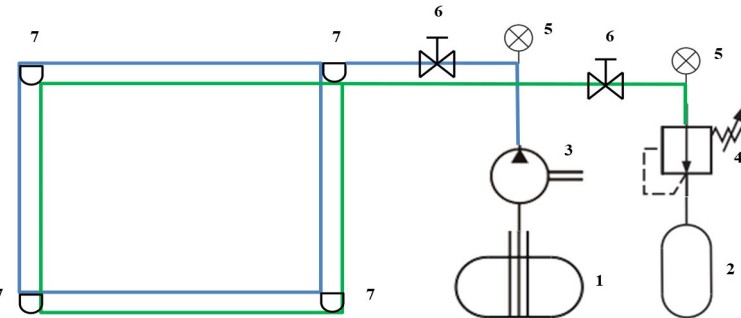

**Figure 2.** Extinguishing system diagram (1—water tank, 2—gas tanks, 3—water pump, 4—gas pressure reducer, 5—manometers, 6—shut-off valves, 7—nozzles).

A network consisting of 26 thermocouples TP 201K-300-15 with a length of 2.5 cm and a diameter of 1 mm was used. The technical data of the thermocouples are included in Table 2.

**Table 2.** Technical data of thermocouple TP 201K-300-15.

| Parameter | Material, Value or Range |
|---|---|
| Measurement range | from −40 °C to + 1150 °C |
| Mantle material | NiCr–NiAl (K) |
| Sheath material | Inconel (T, J, K) |
| Diameter | 0.5 mm |
| Thermocouple length | 300 mm |
| Length of compensation cable | 1.5 m |
| Measurement accuracy | 1.5 °C (−40–375 °C) or 0.4% (375–900 °C) |

Thermocouples included in the measuring grid were arranged to obtain reliable temperature distribution in the room depending on the location of the fire source (corner stack position to obtain the effect of the most unfavorable fire location from the point of view of fire extinguishing). On the ceiling there were ten thermocouples evenly arranged diagonally every 0.5 m. The other thermocouples were placed in four vertical lines (trees) on the walls in the corner of the room. The first of these trees was located at a distance of 0.5 m from the source of the fire and consisted of four thermocouples located, respectively, at heights of: 1.5 m, 2.0 m, 2.5 m and 2.7 m. The second tree, which was also fitted with four thermocouples at the same heights, was located on the adjacent wall, at a distance of 0.285 m from the source of the fire. The third tree, consisting of five measuring points, was located 0.5 m from the fire. In this case, the first thermocouple was located at a height of 0.9 m. The last thermocouple tree was placed 1 m from the fire. It consisted of three thermocouples installed at 0.9 m, 1.5 m and 2.5 m, respectively. The diagram of the arrangement of thermocouples on two adjacent walls and on the ceiling is shown in Figures 3 and 4 (left side). The heights at which the individual thermocouples are mounted are shown in the vertical projection shown in Figure 4 (right side).

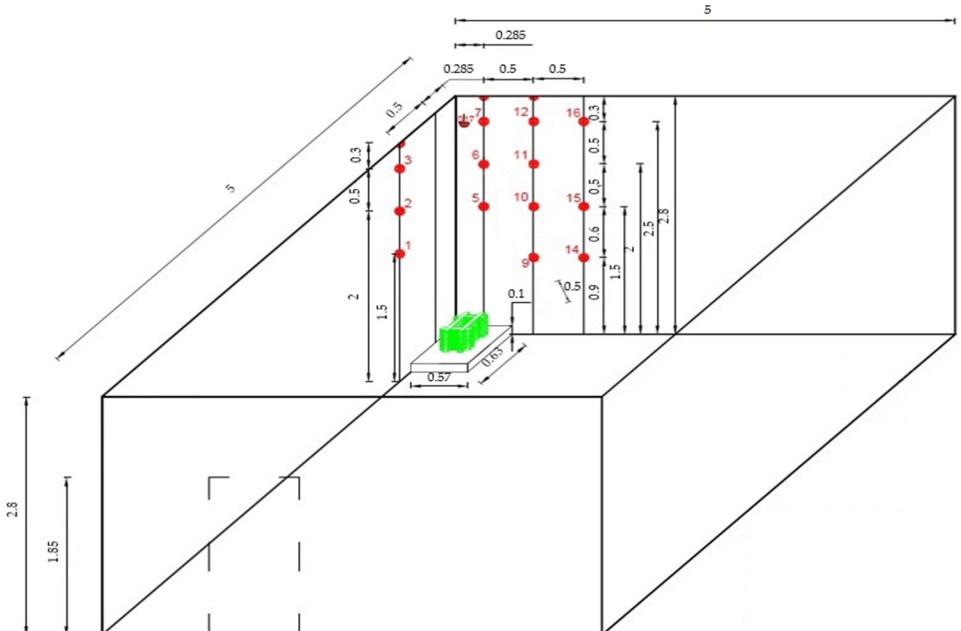

**Figure 3.** Room corner with thermocouples mounted on two adjacent walls.

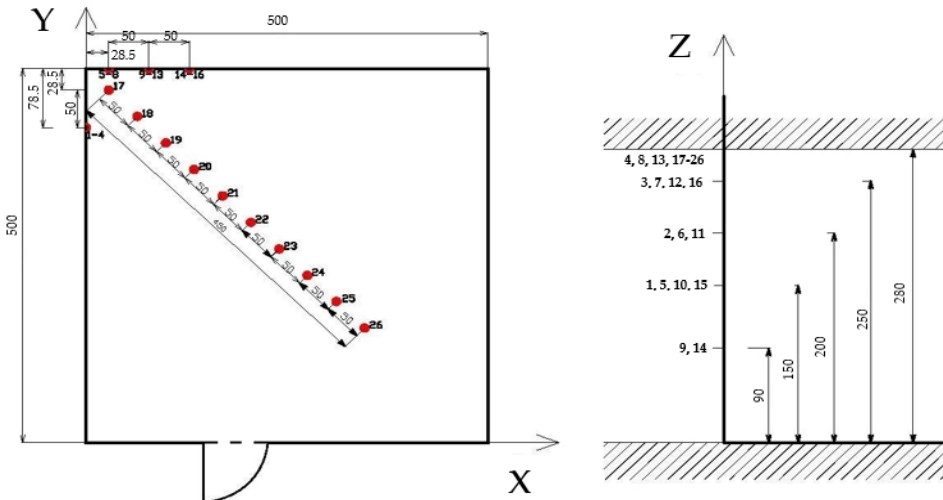

**Figure 4.** Horizontal positions of thermocouples placed under the ceiling (**left**) and vertical positions of the thermocouples on the tree (**right**).

All thermocouples were connected to a computer via Advantech ADAM measuring modules. Each thermocouple number (1–26) corresponded to the number programmed in Advantech ADAMView. Program start and recording of temperature readings began when the wood pile was set on fire. The end of the measurement was unambiguous when the flame of the burning pile was suppressed.

The research was aimed at determining the time of effective extinguishing of flames and comparing the extinguishing efficiency using a combination of water mist and compressed nitrogen as a propellant gas in various proportions. For each test, a pile of wood was created consisting of 50 coniferous boards with a moisture content of 5–15% and a density of about 0.55 kg/m³, which was placed in the corner of the test room. The alternating arrangement of the boards allowed free flow of air to the material being burned and the rapid development of flame combustion between the boards arranged. On the basis of calorimetric tests, an average power of approximately 100 kW was estimated. The pile was placed at a height of about 40 cm above the ground. In order to achieve flame wood burning, about 250 mL of low boiling point kerosene was poured into a metal tray on which the stack was placed. The scheme of stacking and two views of the wooden stack are shown in Figure 5.

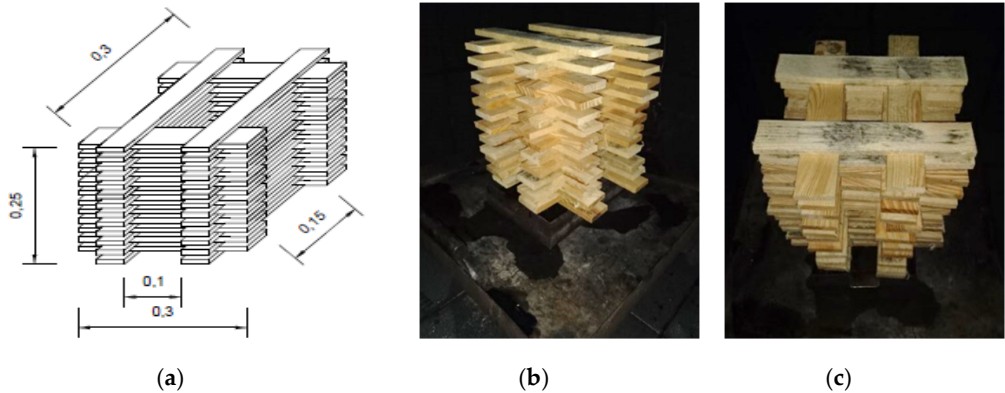

| (a) | (b) | (c) |
|---|---|---|

**Figure 5.** A wooden stack as a flammable material: (**a**) Scheme of stacking; (**b**) side view of the stack; (**c**) top view of the stack.

After ignition, the room door was closed and temperature as well as gas concentrations started to be recorded using a measuring system consisting of sensors, analog–digital converters and a computer with appropriate software. Extinguishing began when the temperature on any thermocouple reached 200 °C (most often it was thermocouple No. 17). The measurement process was terminated when the fire was completely extinguished and the temperature dropped below 150 °C to prevent the flame

returning again spontaneously. The maximum extinguishing time was set at 10 min, assuming the reference times for extinguishing flammable liquids from the FM Global 5580 standard. The room was ventilated and cooled to ambient temperature after each completed test. Some of the criteria for extinguishing effectiveness were the difference between the minimum and maximum temperature and the rate of temperature drop measured by thermocouple No. 17. Due to the location of the thermocouple directly above the combustion zone under the ceiling, temperature changes were best visible on it and its reactions to its changes were the fastest. In addition, it should indicate the maximum temperature values relative to other thermocouples.

Comparison of extinguishing efficiency depending on the volumetric water flow rate is possible with the adoption of appropriate criteria related to temperature curves at various points in the room. During the fire extinguishing process, two-phase mist particles absorb heat generated in the combustion process (receive energy), thanks to which the flame temperature usually drops below the ignition temperature, which significantly reduces the combustion process. The extinguishing effect obtained by lowering the temperature below the flash point is an important advantage of mist installations. The tested system at a volumetric water flow rate of 3.0 $dm^3$/min is according to the standards of this type a system (with the restriction placed on hybrid installations by the NFPA 770 standard), although it differs by a much lower flow than conventional mist installations.

Twenty-three effective extinguishing tests using nitrogen as a propellant gas were carried out. The research was primarily to show that the gas used in the mist extinguishing system, in accordance with the NFPA 750 standard, is not only a driving factor, but also participates in the extinguishing process. The extinguishing process was carried out at four different water mist flow rates 0.5, 1, 1.5 and 3 $dm^3$/min. In addition, extinguishing with pure nitrogen without water mist was carried out in one test, corresponding to flow rate of 0 $dm^3$/min. To systematize the obtained results, the following two points on the time axis and one time interval were distinguished:

- Start—the time from ignition to the start of extinguishing ($t_1$), s;
- Stop—the time from the ignition to the end of extinguishing ($t_2$), s;
- Extinguishing time—$t_g = t_2 - t_1$ — the difference between Stop ($t_2$) and Start ($t_1$), s.

In order to evaluate the test results for all tests carried out in various configurations, the following mathematical criterion of extinguishing effectiveness was introduced:

$$W = 0.5 \cdot W_{tg} + 0.2 \cdot Wv_T + 0.2 \cdot W_{2.0} + 0.1 \cdot W_w, [\%] \tag{1}$$

The following indicators were used in it:

(a) $W_{tg}$ indicator (in the range of 0 to 100 points) specifying the extinguishing efficiency from the point of view of extinguishing time $t_g$, calculated from the following formula:

$$W_{tg} = [1 - (t_g - t_{g\,min})/(t_{g\,max} - t_{g\,min})] \cdot 100, [\%] \tag{2}$$

where: $t_{g\,min}$, $t_{g\,max}$—minimum and maximum values of extinguishing time obtained in all tests, respectively, s.

(b) $W_{VT}$ indicator (in the range of 0 to 100 points) specifying the extinguishing effectiveness from the point of view of the average rate of temperature drop, calculated from the following formula:

$$W_{VT} = [(v_T - v_{T\,min})/(v_{T\,max} - v_{T\,min})] \cdot 100, [\%] \tag{3}$$

where:

$v_{T\,min}$—minimum value of the average temperature drop rate among all compared tests, °C/s;
$v_{T\,max}$—maximum value of the average temperature drop rate among all compared tests, °C/s;
$v_T = (T_{max} - T_{min})/(t_{min} - t_{max})$, °C/s;

$t_{min}$—time to reach maximum temperature ($T_{max}$), s;
$t_{max}$—time to reach minimum temperature ($T_{min}$), s.

A graphic illustration of the maximum and minimum temperature values and times $t_{max}$ and $t_{min}$ is shown in the form of horizontal and vertical blue lines in Figure 6.

(c)　$W_{2.0}$ indicator (in the range of 0 to 100 points) specifying the extinguishing effectiveness from the point of view of the time $t_{60}$ counted from the moment of starting extinguishing to the moment when the average temperature at a height of 2 m drops below 60 °C (critical value), calculated from the following formula:

$$W_{2.0} = [1 - (t_{60} - t_{60\,min})/(t_{60\,max} - t_{60\,min})] \cdot 100, [\%] \tag{4}$$

where:

$t_{60\,min}$—minimum value of time $t_{60}$, s;
$t_{60\,max}$—maximum value of time $t_{60}$, s.

(d)　$W_w$ indicator (in the range of 0 to 100 points) specifying the extinguishing effectiveness from point of view of the amount of water used during extinguishing, calculated from the following formula:

$$W_w = [1 - (V_w - V_{w\,min})/(V_{w\,max} - V_{w\,min})] \cdot 100, [\%] \tag{5}$$

where:

$V_{w\,min}$—minimum value of water consumption among all compared tests, $dm^3$;
$V_{w\,max}$—maximum value of water consumption among all compared tests, $dm^3$;
$V_w$—water consumption in the selected test, $dm^3$.

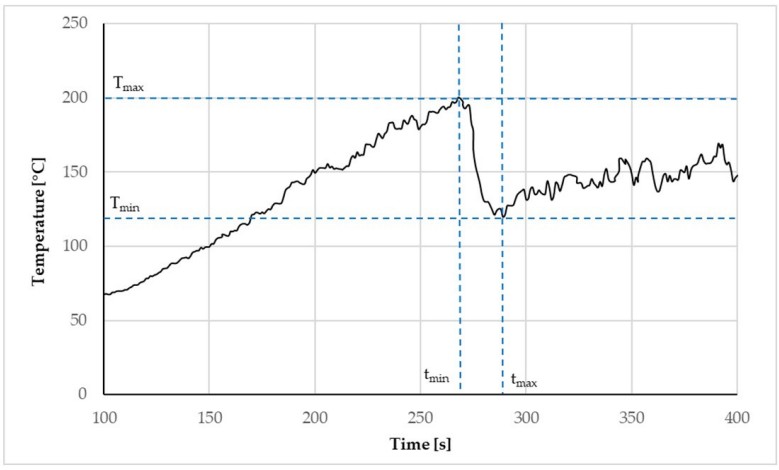

**Figure 6.** Temperature curve with marked values of $T_{min}$, $T_{max}$, $t_{min}$ and $t_{max}$.

The proposed indicators represent firefighting efficiency criteria due to different parameters and enable us to compare the results obtained in individual tests with each other.

The analysis assumed different weights of individual components. The most important criterion, represented by the $W_{tg}$ indicator, was the time of extinguishing, for which a weighting factor equal to 0.5 was adopted. From the point of view of fixed firefighting efficiency, this indicator is the most important. Shortening the time of exposure to fire and heat radiation (both the room and the equipment it contains) reduces fire losses, as well as the emission of toxic combustion products to the atmosphere. This parameter is commonly used in the evaluation of extinguishing systems in certification tests. The less important parameters were those described by indicators $W_{vT}$ (average temperature drop

rate) and $W_{2.0}$ (connected with time $t_{60}$), for which weighting factors equal to 0.2 were adopted. The indicator $W_w$, related to the water consumption, was assumed as the least significant from the point of view of firefighting efficiency. Thus, a weighting factor of 0.1 was assumed for this criterion. In calculating the final total of points, appropriate weights were used for individual partial criteria, according to the following formula:

The final sum of points obtained was converted into a percentage on a scale from 0 to 100% according to the following formula:

$$P = [(W - W_{min})/(W_{max} - W_{min})]\cdot 100, [\%] \tag{6}$$

where:

W—total points including partial criteria calculated according to Formula (1);
$W_{min}$, $W_{max}$—minimum and maximum points, respectively.

To analyze the impact of volumetric flow rate and gas type on the average temperature at two different altitudes, 2.0 m and 1.5 m, significant from the point of view of human evacuation, they were calculated using the following formulas:

$$T_{a\,2.0} = (T_2 + T_6 + T_{11})/3, [°C] \tag{7}$$

$$T_{a\,1.5} = (T_1 + T_5 + T_{10} + T_{15})/4, [°C] \tag{8}$$

where $T_i$ is the temperature measured by the i-th thermocouple (see Figures 3 and 4).

## 3. Results

The time values, defined in Section 2, obtained during all tests carried out at different water flow rates, are presented in Table 3. The last column contains the deviation from the mean value of extinguishing time calculated for tests in which the same water flow rate was used.

**Table 3.** The time parameters (start, stop, extinguishing time) and deviation from the mean value of extinguishing time for all positive extinguishing tests at different water flow rates.

| No. of the Test | Water Flow Rate (dm³/min) | Start (s) | Stop (s) | Extinguishing Time (s) | Deviation From the Mean Value of Extinguishing Time |
|---|---|---|---|---|---|
| 1 | 0.0 | 268 | 620 | 352 | 0 |
| 2 | 0.5 | 328 | 634 | 306 | 13 |
| 3 | 0.5 | 254 | 534 | 280 | 7 |
| 4 | 1.0 | 303 | 526 | 223 | 57 |
| 5 | 1.0 | 269 | 553 | 284 | 4 |
| 6 | 1.0 | 320 | 611 | 291 | 11 |
| 7 | 1.0 | 327 | 613 | 286 | 6 |
| 8 | 1.0 | 240 | 528 | 288 | 8 |
| 9 | 1.0 | 308 | 618 | 310 | 30 |
| 10 | 1.5 | 260 | 457 | 197 | 16 |
| 11 | 1.5 | 269 | 463 | 194 | 19 |
| 12 | 1.5 | 217 | 441 | 224 | 11 |
| 13 | 1.5 | 206 | 432 | 226 | 13 |
| 14 | 1.5 | 351 | 566 | 215 | 2 |
| 15 | 1.5 | 283 | 508 | 225 | 12 |
| 16 | 3.0 | 387 | 580 | 193 | 5 |
| 17 | 3.0 | 330 | 570 | 240 | 42 |
| 18 | 3.0 | 201 | 390 | 189 | 9 |
| 19 | 3.0 | 265 | 470 | 205 | 7 |
| 20 | 3.0 | 324 | 530 | 206 | 8 |
| 21 | 3.0 | 243 | 453 | 210 | 12 |
| 22 | 3.0 | 273 | 430 | 157 | 41 |
| 23 | 3.0 | 325 | 510 | 185 | 13 |

Due to the lack of repeatability of fire development, despite the efforts to ensure the same conditions and, in many cases, too few tests, detailed statistical analysis was omitted. When choosing a test for a given water flow rate. which was further analyzed, the value of deviation of extinguishing time from the average value was taken into account (see Table 3).

Temperature changes over time were best seen on thermocouple No. 17 mounted on the ceiling, directly above the location of the test fire. Its reaction to extinguishing is the fastest, because it has the highest sensitivity to changes occurring in the combustion zone. Therefore, for comparative analysis of temperature, the temperature values measured by this thermocouple were adopted. Temperature time series measured by this thermocouple for various water mist flow rates are presented in Figure 7a. The most important fragments of the graphs covering the time immediately before and after the beginning of extinguishing are shown in Figure 7b. Vertical dashed lines mark the moment of switching on the fire extinguishing system. The average temperature calculated on the basis of temperatures measured by thermocouples No. 1, 5, 10 and 15 placed at a height of 1.5 m and thermocouples No. 2, 6 and 11 placed at a height of 2 m during tests carried out at different values of water flow rates are presented in Figure 8a,b, respectively.

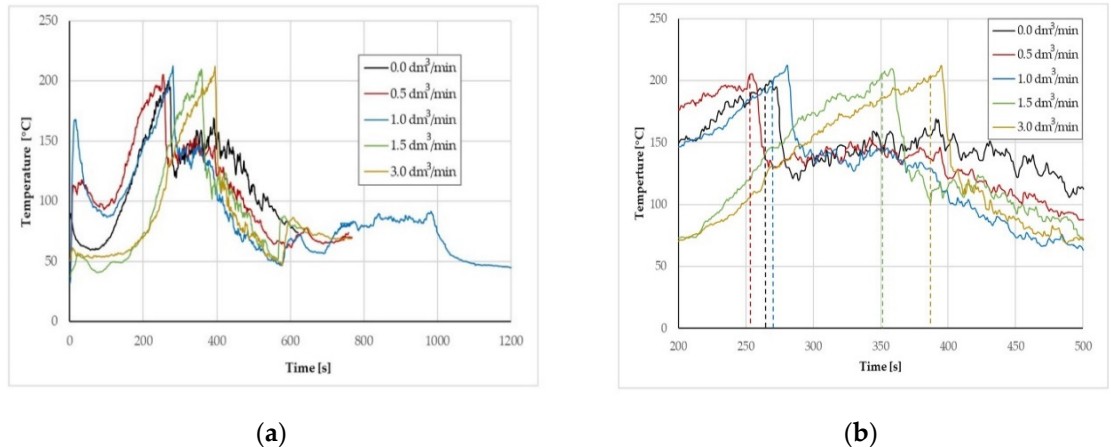

(**a**)            (**b**)

**Figure 7.** Temperature measured by thermocouple No. 17 at different water flow rates: (**a**) During whole fire test; (**b**) between 200 and 500 s.

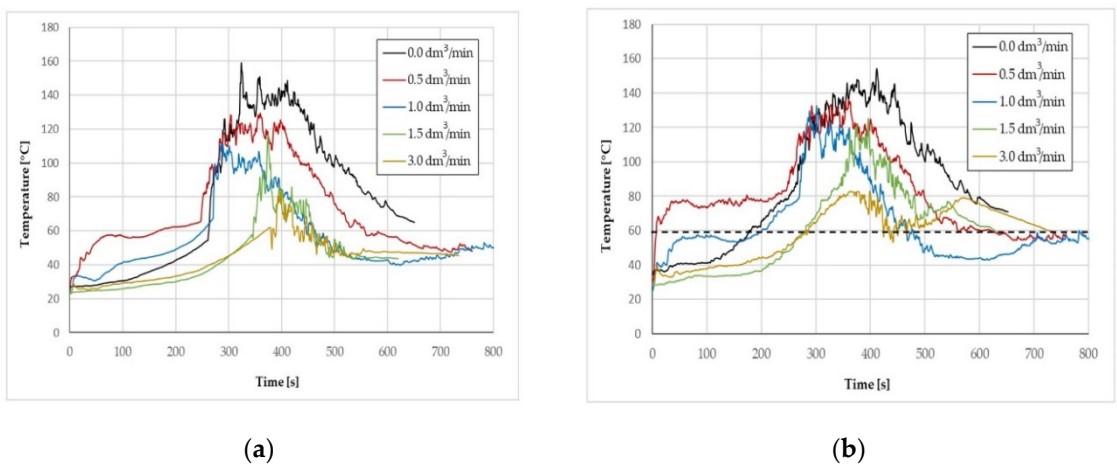

(**a**)            (**b**)

**Figure 8.** Average temperature calculated on the basis of temperatures measured during fire tests at different water flow rates by thermocouples mounted at a height of: (**a**) 1.5 m (No. 1, 5, 10 and 15); (**b**) 2.0 m (No. 2, 6 and 11).

The test with the lowest value of deviation was chosen. Values of parameters defined in Section 2 and accepted as an extinguishing efficiency criteria for selected tests are included in Tables 4 and 5.

**Table 4.** The time parameters (start, stop, extinguishing time), $W_{tg}$ indicator, water consumption and standard deviation of extinguishing time for selected extinguishing tests at different water flow rates.

| No. of the Test | Water Flow Rate (dm³/min) | Start (s) | Stop (s) | Extinguishing Time (s) | $W_{tg}$ (%) | Water Consumption (dm³) | Standard Deviation of Extinguishing Time |
|---|---|---|---|---|---|---|---|
| 1 | 0.0 | 268 | 620 | 352 | 0 | 0.00 | 0 |
| 3 | 0.5 | 254 | 534 | 280 | 45 | 2.33 | 13 |
| 5 | 1.0 | 269 | 553 | 284 | 43 | 4.73 | 27 |
| 14 | 1.5 | 351 | 566 | 215 | 86 | 5.37 | 13 |
| 16 | 3.0 | 387 | 580 | 193 | 100 | 9.65 | 26 |

**Table 5.** Values of parameters accepted as extinguishing efficiency criteria for selected tests.

| No. | No. of Test | Water Flow Rate (dm³/min) | $T_0$ (°C) | $v_T$ (°C/s) | $W_{VT}$ (%) | $t_{60}$ (s) | $W_{2.0}$ (%) | $W_w$ (%) | $W$ (%) | $P$ (%) |
|---|---|---|---|---|---|---|---|---|---|---|
| 1 | 1 | 0.0 | 81 | 3.8 | 4 | ** | 0 | 100 | 10.8 | 0 |
| 2 | 3 | 0.5 | 75 | 5.8 | 57 | 307 | 27 | 76 | 46.9 | 61 |
| 3 | 5 | 1.0 | 79 | 3.8 | 4 | 198 | 100 | 51 | 47.4 | 62 |
| 4 | 14 | 1.5 | 108 | 3.7 | 0 | 278 | 46 | 44 | 56.6 | 77 |
| 5 | 16 | 3.0 | 102 | 6.0 | 100 | 347(48) * | 0 | 0 | 70.0 | 100 |

\* For the first time, the average temperature dropped to 60 °C after about 48 s from the beginning of extinguishing, however, due to its re-increase to almost 80 °C, a longer period was taken into account, after which the average temperature dropped below 60 °C (347 s) for the second time and this tendency persisted until the end of the test.
\*\* The average temperature at a height of 2 m did not fall below 60 °C until the end of the test. The value of $W_{2.0} = 0$ was adopted in this case.

Selected fragments of the temperature curves are shown in Figure 7 just before and after the start of extinguishing.

## 4. Discussion

The discussion of the results obtained, aimed primarily at demonstrating the impact of water flow rate on the extinguishing efficiency of the analyzed hybrid system, is based on the temperature charts shown in Figures 7 and 8 and the values of parameters and indicators collected in Tables 3–5.

The extinguishing time was considered to be the most important parameter proving the extinguishing effectiveness, which was the longest ($t_g$ = 352 s) during extinguishing with pure nitrogen without the participation of water mist and the shortest ($t_g$ = 193 s) during extinguishing with the use of water mist fed with a flow rate of 3 dm³/min. In no case, however, did it exceed the maximum time of 8 min, in accordance with current requirements. A slightly longer extinguishing time ($t_g$ = 215 s) was obtained in the case of a hybrid extinguishing system in which the mist flow rate was equal to 1.5 dm³/min. Despite the fact that almost half the amount of water was used at that time (9.65 dm³ at a flow rate of 3 dm³/min and 5.37 dm³ at a flow rate of 1.5 dm³/min), however, due to its low cost, a shorter extinguishing time should be considered as decisive when choosing the extinguishing variant.

The second most important parameter, according to which the rate of temperature drop after the beginning of extinguishing was assessed, in this case also turned out to be the highest (6 °C/s) with a flow rate of 3 dm³/min. A comparable value was obtained for a flow rate of 0.5 dm³/min (5.8 °C/s), while in other cases, it was much smaller and equal to about 3.8 °C/s.

Another parameter compared was the temperature difference between the maximum and minimum values. Clearly, higher values were obtained for the largest water capacities used, 1.5 dm³/min (108 °C) and 3.0 dm³/min (102 °C). For the remaining cases, this difference is comparable and amounts to about 80 °C. As for the last indicator taken into account, defined as the time after which the average temperature falls lower than 60 °C at a height of 2 m, it is taken as critical and, despite this, the fastest decrease occurring during extinguishing with a flow rate of 3.0 dm³/min (after 48 min) and, due to a further increase, it exceeded this limit, reaching it again only after 347 s (this time was taken to calculate the indicator, because it is important to keep the temperature below this level throughout the test).

Considering the value of the global extinguishing efficiency index corresponding to the assumed criterion in the form of Formula (1), which takes into account all of the above the indicators, shows that its value gradually increases as the water flow rate through the nozzle increases. The biggest difference occurs between the value of the indicator obtained during extinguishing with pure nitrogen when about four times more gas was used than in other cases (W = 2.6%) and that obtained during extinguishing even with a minimum amount of water, 0.5 dm$^3$/min (W = 39.3%). We have a similar difference between the global indicator values, about 30%, when extinguishing a fire with a minimum (1.5 dm$^3$/min) and maximum water efficiency (3 dm$^3$/min).

Comparing the obtained results, especially extinguishing times, with the results obtained during the suppression with three different mist nozzles (without the use of inert gas) described by Gałaj and Drzymała in 2018, it can be stated that they were on average half the length in the case of the hybrid system, despite the fact that extinguishing started at a higher temperature (200 °C for the hybrid system and 150 °C for the classic mist system) [17].

Summing up the conducted research and in particular the analysis of the results obtained, it can be stated that the most effective extinguishing system turned out to be a hybrid system using a nitrogen-powered water mist fed with a flow rate of 3 dm$^3$/min. In this case, only about 10 dm$^3$ of water was consumed during the entire extinguishing, which is significantly less than in the case of conventional sprinkler systems and even standard mist systems, which also significantly reduces fire losses. Future research by the authors is planned, consisting in an analysis of the selection of optimal gas and water ratios depending on the fire power as well as the size of the room and its tightness.

**Author Contributions:** Conceptualization, P.W., J.G. and T.D.; methodology, P.W., J.G. and T.D.; formal analysis, J.G., N.T., P.W. and T.D.; investigation, P.W. and J.G.; resources, P.W., N.T. and J.G.; data curation, P.W., N.T. and J.G.; writing—original draft preparation, J.G., T.D. and N.T.; writing—review and editing, J.G., T.D. and N.T.; visualization, J.G. and N.T.; supervision, J.G. and P.W.; project administration, T.D. and J.G.; funding acquisition, P.W. All authors have read and agreed to the published version of the manuscript.

**Funding:** Statutory project financed from means of the Minister of Science and Higher Education for the Main School of Fire Service No. S/E-422/31/17/18, entitled: "Analysis of Extinguishing Effectiveness of Sprayed Jets Generated by Selected Mist Nozzles, Mist Nozzles Type Turbo and Extinguishing Lances During the Extinguishing of Type A Fires in a Closed Premise".

**Conflicts of Interest:** The authors declare no conflict of interest, the funders had no role in the design of the study; in the collection, analyses, or interpretation of data; in the writing of the manuscript, or in the decision to publish the results.

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
