# Peer review of "Analysis of the Impact of Water Flow Rate on the Temperature Variability in a Closed Room during the Extinguishing of A-Group Fire Using a Hybrid Water Mist Suppression System"

_sustainability, doi:10.3390/su12208700_

Round 1

Reviewer 1 Report

With some modifications, the paper will be suitable for submission to MDPI's Fire journal so that appropriate suggestions for improvement can be obtained. There are some edits that this reviewer has provided in the highlighted parts of the attached "pdf" file.

Reviewer 2 Report

In general, some involvement of inert gases in water mist extinguishing in the extinguishing process is assumed. It can also be expected that the best extinguishing results will be achieved with the largest water supply. In this sense, the article does not present new information. The experiments only confirm the above facts.

Reviewer 3 Report

The following comments will be based on the line number in the manuscript.

Line 34 - An introduction to the topic would be beneficial to provide context as to why this is important.

Line 35 - It seems like a citation is needed here. In addition, the next sentence about the US Navy doing research is not related to anything yet since there was noting cited.

Line 35 - What effect?  You write: "Their effect.."

Line 39 - It would help to describe/define "hybrid systems".

Line 45 - You write "only a few works.." What works?  Are their citations?

Line 47 - The end of the sentence does not make sense - .. found at work [5]? What is this?

Line 47 - What is FM Global?

Lines 49 through 60 are very difficult to follow.  Is this section needed?

Lines 70-71 - How can something be theoretically used to provide protection?

Lines 72-74 - Very long sentence that is hard to follow what the thought is.

Lines 82-83 - Difficult to understand this sentence.

Lines 88-89 - It seems more explanation is needed of something that is purely a theoretical possibility if it hasn't been implemented.

Line 90 - You say "Based on research .." What research?

Lines 90-92 - A very long sentence that is hard to follow.

Line 93 - Should there be a citation for NFPA 2001 and 750?

Lines 188-189 - The sentence does not make sense.

Line 197 - Does there need to be a citation for the Laval nozzle?

Line 200 - Do you need to explain "a theoretical Mach number of 2"?

Line 208 - The sentence needs to be rewritten based on the citation included.

Lines 296-298 - It is hard to follow the thought in these 2 sentences.

Lines 301-303 - This is a 1-sentence paragraph that is hard to follow.

Line 304 - Do you define Wtg?

Line 323 - Figure 6 needs a more descriptive title. 

Lines 345-347 - The sentence is hard to follow.

Lines 351-356 - Very hard to follow this section.

Line 367 - "A twenty three.."?  Should that be 23?

Lines 366-376 - This seems like it belongs in the Methods section and is not a Result.

Line 398 - You mention Table 5 before you mention Table 4. 

Line 425 - Is there a Figure 9?

Lines 423-454 - This is a very long paragraph that covers a few topics and is hard to follow.

Lines 455-456 - What previous ones?

Line 462 - Provide the author(s) for citation [17].

Lines 471-472 - A one sentence paragraph.

Do you have any more concluding remarks?

Round 2

Reviewer 1 Report

This reviewer appreciates the efforts to revise the manuscript. However, it is still problematic that the theme of sustainability is not reflected in this manuscript. An objective reviewer or editor would likely regard it as being more suitable for MDPI's "Fire" or fire-related journals of other publishers based on the editorial guidelines of "Fire" versus "sustainability". It is to the advantage of the authors to get this manuscript accepted at a journal with the appropriate thematic scope in order to get better citation statistics. This reviewer does not follow the reasoning why the related 2019 article being published in "sustainability" and the Ministry of Education's "journal point system" should influence the decision to select this journal while there is a more suitable journal for this manuscript. This reviewer sincerely believes that there are issues of sustainability that can be discussed for the hybrid water mist fire suppression system in comparison to the harmful perfluoro-based fire suppressants. 

The other issue that needs to be addressed is the frequent reference to BP Global including the highlighted verbatim quote of company's business focus in the statement quoted below. This seems like unnecessary promotion of a commercial entity and should be rephrased to focus on the FM Approval 5560 and 5580 standards.

"The tests carried out by FM Global (American mutual insurance company based in Johnston, Rhode Island, United States, with offices worldwide, that specializes in loss prevention services primarily to large corporations throughout the world in

the Highly Protected Risk (HPR) property insurance market sector)....."

Below are two weblink to help the authors understand the need to avoid repetitive mentioning of company names or products so that it does not sound like advertisement or promotion. This should be easy for the authors to avoid explicitly mentioning "FM Global" by citing the company's standard documents and discussing the specifics of the standards in the narrative. 

https://www.ncbi.nlm.nih.gov/pmc/articles/PMC2735431/ 

https://www.ncbi.nlm.nih.gov/pmc/articles/PMC3035881/ 

Author Response

Dear Reviewer,

Adding the description of FM Global was a suggestion of one of the reviewers. Due to your suggestion, we removed the text in parentheses, adding at the same time before the name with which it should be associated. In addition, we tried to limit its administration to the necessary minimum.

Regarding the revision suggestion, however, we will insist that our article be published in journal “Sustainability” for the following reasons:

  1. a) compliance of the subject of the article with regard to sustainable development is related to at least three issues: water saving due to the use of its atomized form (water mist), the use of nitrogen as an inert gas that is present in the air anyway, no need to use halon (harmful to the environment) or its substitutes for extinguishing.
  2. b) acceptance by the editors of the article for review (if it did not match the profile of the journal, it would be rejected at the very beginning),
  3. c) application of a special number "Green Technologies in Air Treatment" by our institution,
  4. d) lack of the suggested magazine "Fire" in the list of scoring journals of the Ministry of Science and Higher Education, which excludes the recognition of the paper for our scientific achievements and consent to pay for the publication in this journal.

Best regards

Jerzy Gałaj

Norbert Tuśnio

Paweł Wolny

Tomasz Drzymała

Reviewer 3 Report

You did a good job addressing comments - thanks

Author Response

Dear Reviewer,

Thank you for taking the time to review our article in detail and accepting it for publication after the corrections have been made.

Best regards

Jerzy Gałaj

Norbert Tuśnio

Paweł Wolny

Tomasz Drzymała